# Al/Niobium Diboride Nanocomposite's Effect on the Portevin-Le Chatelier Phenomenon in Al-Mg Alloys

**David Florián-Algarín [1]** , **Michelle Marrero-García [2]**, **Javier José Martínez-Santos [3]**,
**Luis Montejo Valencia [4]** and **Oscar Marcelo Suárez [4],***

[1]   Department of Civil Engineering, University of Puerto Rico-Mayagüez, Mayagüez, PR 00681, USA
[2]   Department of Mechanical Engineering, University of Puerto Rico-Mayagüez, Mayagüez, PR 00681, USA
[3]   Department of Mechanical Engineering, University of Wisconsin-Madison, Madison, WI 53706-1607, USA
[4]   Department of Engineering Science and Materials, University of Puerto Rico-Mayagüez, Mayagüez,
     PR 00681, USA
*   Correspondence: oscarmarcelo.suarez@upr.edu

**Abstract:** In Al-Mg alloys, the Portevin-Le Chatelier phenomenon, or dynamic strain aging, reveals itself as serrations upon plastic tensile deformation. This research evaluates this phenomenon when Al/NbB$_2$ nanocomposite pellets are added to a magnesium-supersaturated Al matrix. A ball-milled 90 wt % Al and 10 wt % NbB$_2$ nanocomposite helped inoculate an Al-Mg melt to incorporate the nanoparticles effectively. The melt was cast into rods that were cold-rolled into 1 mm diameter wires. Two sets were prepared: The first group was an as-cast set of samples, for comparison purposes, whereas the second was a solution-treated set. The solution treatment consisted of annealing followed by ice-water quenching. The results corroborating that the phenomenon was observable only in the specimens bearing the solution treatment, were used as the research baseline. Said treated alloy was compared to one containing the nanoparticles, which proved that the NbB$_2$ particles caused a reduction of the serrated signal amplitude.

**Keywords:** Portevin-Le Chatelier phenomenon; Al-Mg; nanocomposite; Fourier transform

## 1. Introduction

In the manufacturing of light vehicles with Al-Mg alloys, one relevant challenge is to keep automotive and aerospace parts at reasonable prices [1–3]. In this strict context, a problem limiting the use of Al-Mg alloys is the dynamic strain aging occurring at room temperature when these alloys are cold-formed [4]. This phenomenon, discovered by Portevin and Le Chatelier in 1909 [5,6], affects the alloys' flow stress, ultimate tensile strength, work hardening rate, and their ductility [2,3,7]. Additionally, the affected alloys display increased embrittlement and a reduction in fracture toughness [1]. This visible jerky plastic flow present in the alloys' tensile stress-strain curves was to be known later as the Portevin-Le Chatelier (PLC) phenomenon. Further studies led to the conclusion that the serrations were the result of the plastic instabilities in the material [1,3]. Currently, the accepted explanation for the occurrence of the PLC effect in Al-Mg alloys is the interaction between the dislocations moving through the material upon the plastic flow and solute atoms in solid solution [1–3,5,8]. Also known as dynamic strain aging (DSA), PLC occurs upon plastic deformation of the alloy with moving dislocations encountering the diffusing substitutional solute atoms [1–3].

At this point, one must underscore that most investigations on the Portevin-Le Chatelier phenomenon have focused on two aspects. The first one has been to determine the variables that affect the PLC phenomenon. Strain rate, specimen roughness, microstructures of the alloys, and Mg content, are among the most studied variables. The second aspect has been the development of

numerical models that describe the PLC phenomenon, like those developed by Lasko, Jiang, and Hu, among others [9–12].

It is well known that, as plastic deformation generates dislocations that can hinder the movement of other dislocations, this phenomenon can decrease the amplitude of the serration signal [13]. Further, grain refined Al-Mg alloys have shown to suppress the PLC effect [13,14].

The Mg content can also affect the amplitude of the stress serration in the PLC phenomenon. Additionally, the critical strain rate varies with the Mg content because higher Mg levels lead to higher critical strain rates; however, this does not have any apparent effect on the width of the PLC bands [15]. The strain rate changes the serrations types [4,15]. For instance, in an Al–3 Mg wt % alloy, type A, B, and C oscillations occur when the strain rates are $10^{-3}$/s, $10^{-4}$/s, and $10^{-5}$/s, respectively [4].

In summary, the reported literature has focused on determining the variables that affect the PLC phenomenon, via either experimental investigations or numerical modeling. Following that trend, the present work proposes a similar methodology applied to nano-strengthened Al alloys, similar to ones developed in previous research [16–19]. Accordingly, we evaluated the effect of $NbB_2$ nanoparticles (as constituents of Al/$NbB_2$ nanocomposite pellets) on the Portevin-Le Chatelier phenomenon in solution-treated aluminum-magnesium alloys. To the authors' knowledge, this is the first research assessing DSA in Al-Mg-$NbB_2$ via MATLAB™ and statistical analysis, to qualify and quantify the PLC signal as a function of the chemical composition, and amount of nanoparticles present in the material. Moreover, this study addresses the potential effect of nanoparticles on plasticity and mechanical strength when the PLC phenomenon affects said alloys.

## 2. Sample Fabrication Methodology

$NbB_2$ pieces (Alfa-Aesar, Ward Haverhill, MA, USA) were fractured to obtain the nanoparticles using a vario-planetary high-energy ball mill (Pulverisette 4, Fritsch GmbH, Idar-Oberstein, Germany). After 10 milling hours at 1600 rpm, using Scherrer's equation, the diboride particle size was estimated to be 17 nm, according to a methodology developed in prior research [17–19]. Promptly, we mixed the nanoparticles with aluminum powder (Acros Organics, Morris Plains, NJ, USA) in the same ball mill to produce Al/$NbB_2$ nanocomposite pellets by cold welding, according to a technique fine-tuned in prior research [17–19]. The so devised nanocomposite pellets (90 wt % Al and 10 wt % $MgB_2$) were sintered at 260 °C for 30 min to enhance the boride-aluminum interface [17–20].

Thence, we prepared a Al–1 wt % Mg molten alloy by diluting a Al–10 wt % Mg master alloy with pure aluminum. We added the nanocomposite pellets into the melt to form 6 mm diameter ingots; the target levels of $NbB_2$ were: 0.0, 0.5, 1.0, and 1.5 wt %. Four specimens of each composition were manufactured. Subsequently, these ingots were cold rolled to obtain 1.0 mm diameter wires; i.e., a cross-sectional area reduction of 97%.

A Nikon Epiphot 200® optical microscope allowed inspecting and imaging the wires' microstructure. With the described procedure, we manufactured two sample sets: While one group of wires underwent the solution treatment for 30 min at 300 °C and ice-water quenching, the other group (as-cold formed) was not heat treated. We completed the standard tensile tests at a $3.3 \times 10^{-5}$/s (i.e., $3.3 \times 10^{-5}$ mm/mm/s) strain rate in an Instron® 5944 low force universal testing machine, according to ASTM B557-06 [21].

## 3. Results and Discussion

As mentioned previously, after inoculating the melt with the nanocomposite pellets we cold-formed the ingots bearing the nanoparticles. Said cold rolling allowed preparing 1 mm diameter wires for the tensile tests. We used the stress–time data (concurrently collected with the stress–strain one) for MATLAB™ and statistical analyses. The ensuing section presents a full analysis of the results.

### 3.1. Fabrication of Al/NbB$_2$ Nanocomposite Pellets

The NbB$_2$ nanoparticles and aluminum powder (with an average size of 44 µm, and 99.5% purity) were mixed at 1000 rpm for 1 h in the vario-planetary ball mill, to form the Al/NbB$_2$ pellets via cold welding. Figure 1 presents the microstructure of this Al/NbB$_2$ nanocomposite pellet. The optical image demonstrates how appropriate insertion of the nanoparticles in the aluminum matrix was attained.

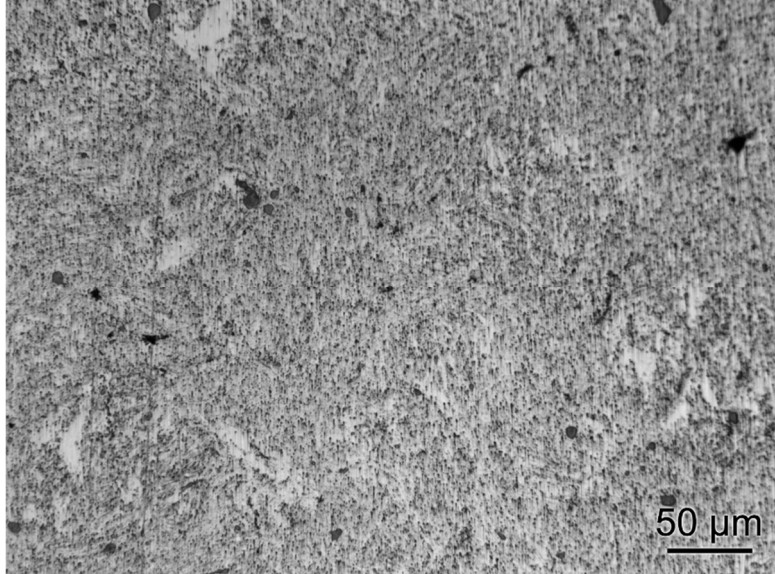

**Figure 1.** Low magnification optical micrograph of the nanocomposite pellets.

### 3.2. Optical Micrographs of Wires

Figure 2a,b presents the microstructure of the samples at different stages of the manufacturing process. Figure 2a shows the grain structure of the as-cast Al–1 wt % Mg inoculated with 1 wt % NbB$_2$. In Figure 2b, one can observe the cold-worked grains after a 97% cross-area reduction of the aluminum wires.

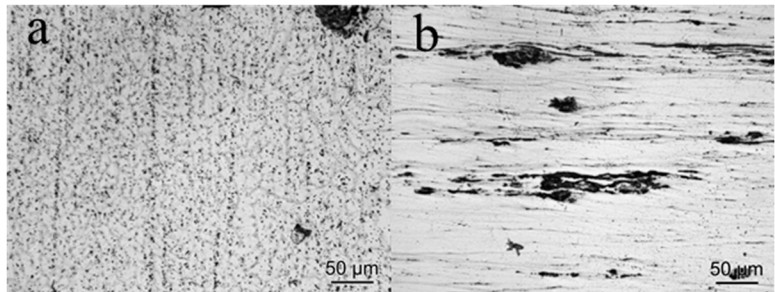

**Figure 2.** Microstructures of the wires at different stages of the manufacturing process: (**a**) Al–1 wt % Mg ingot treated with 1 wt % of NbB$_2$ nanoparticles; (**b**) final sample with Al–1 wt % Mg–1 wt % NbB$_2$ and 1 mm diameter.

### 3.3. Tensile Test Curves

Figure 3 presents examples of the resulting tensile curves as stress versus strain. As expected, the wires without the solution treatment do not show the Portevin-Le Chatelier phenomenon, because the Mg atoms are not in solid solution due to the thermodynamic stabilization of the microstructure. Similar behavior has been observed in a 5083 aluminum alloy sheet, where only the annealed alloys presented the PLC effect [22].

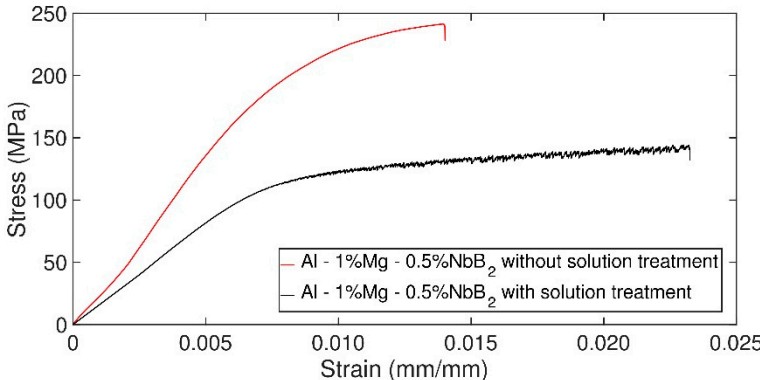

**Figure 3.** Tensile test curve of the aluminum wires with (red curve) and without solution treatment (black curve).

### 3.4. Numerical Analysis of the PLC Signal

Figure 4 presents examples of the resulting tensile curves as stress versus time (rather than stress vs. strain), after removing the elastic region of the material. Using MATLAB™, we cut off the elastic segment (or region of the curve) using the 0.2% strain offset that defines the proof stress [23].

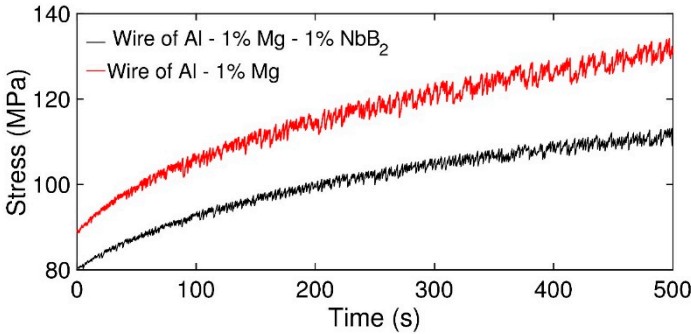

**Figure 4.** Tensile curve (tension versus time) displaying the PLC phenomenon.

Using Fourier transformation, the frequency components were separated in the time domain; that is, the PLC signal, and the environmental noise or vibrations of the equipment, among others. Pure aluminum samples were manufactured to undergo tensile tests with identical parameters as the ones used on the Al–Mg–NbB$_2$ specimens. The results from the pure aluminum samples were analyzed using Fourier transformation; the ensuing tensile curve signals only revealed vibrations for frequencies below 0.2 Hz. Thus, these vibrations were attributed to the noise of the tensile test instrument.

Moreover, all solution-treated samples, i.e., containing Mg, showed vibration frequencies below 5 Hz. Hence, we attributed all frequencies larger than 0.2 Hz to the Portevin-Le Chatelier phenomenon. Frequencies smaller than 0.2 Hz were set equal to zero to construct the corrected stress vs. time curves (filtered stress). Figure 5 displays both curves, i.e., the as-acquired (original) curve and the filtered one. Upon the analysis of the PLC effect, the filtered curve subtracted from the original curve, to obtain the PLC signal, which is presented in Figure 6.

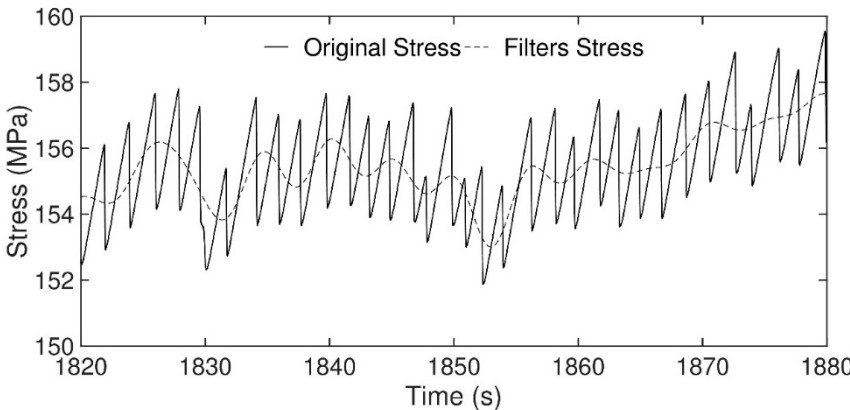

**Figure 5.** The original stress-time curve obtained from solution-treated Al–1 wt % Mg wires and the new filtered signal after conversion to its time domain.

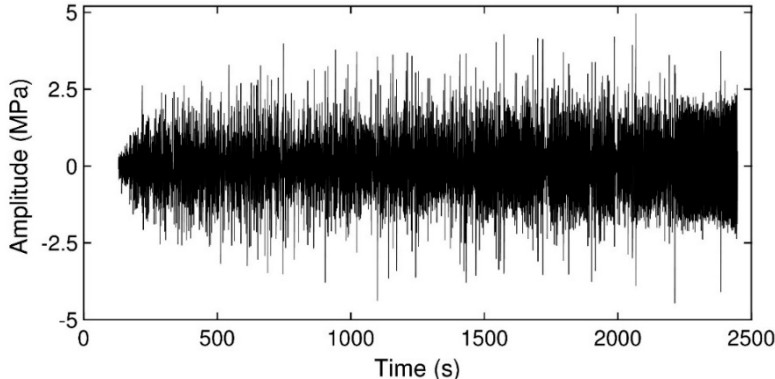

**Figure 6.** New stress-time curve after subtracting the filtered signal.

Then, to transform all positive values of the PLC signal, one needs to calculate the area under the peaks by numerical integration. Thus, the absolute value of the PLC signal was computed according to Figure 7. The value of the areas below the peaks was then divided over time to normalize the stress vector. The function used in the code computed the integral via the trapezoid method, for the sake of simplicity. For analysis purposes, we were interested in the value of the entire sum, which, when divided by the total time, gave a normalized value of stress caused by the PLC effect.

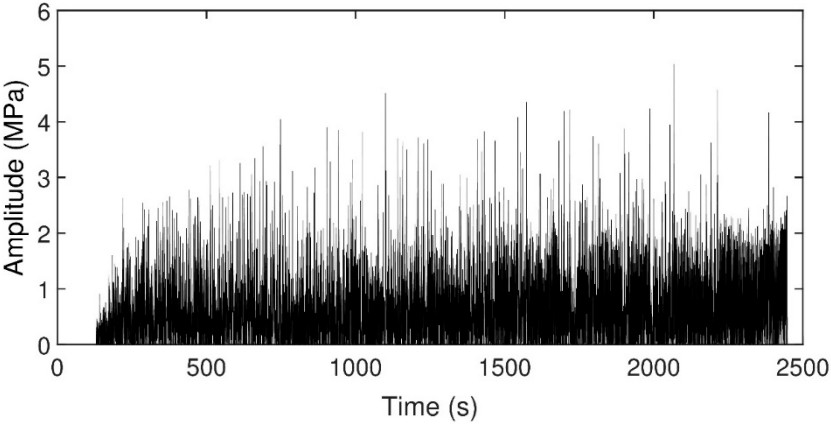

**Figure 7.** The absolute value of stress plotted against time.

The amplitude of the PLC signal in Figure 8 demonstrates how more nanoparticles can lower the amplitude of the phenomenon. Therefore, one can infer that the $NbB_2$ nanoparticles partially hinder the precipitation of Mg-containing metastable phases from the aluminum solution solid. The revised literature provides examples with similar results in AA2017 alloys with SiC particles [24]. The authors attributed the decrease in the serration amplitude in the SiC-reinforced alloys to "the stress fields of the particles, which represent obstacles for dislocation motion and thus increase the pinning effect" [24].

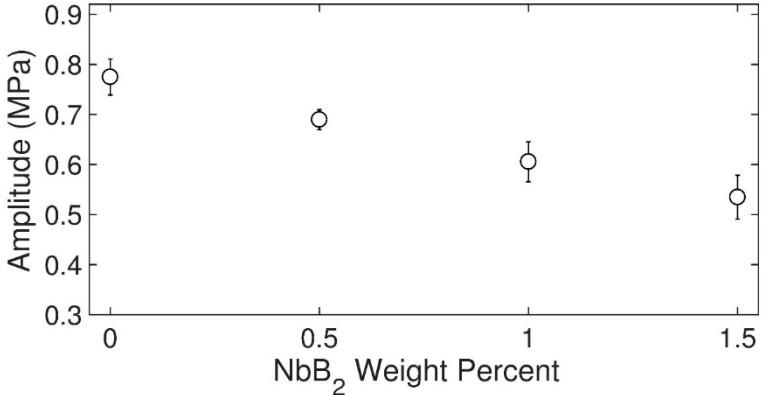

**Figure 8.** The amplitude of signal PLC in solution-treated alloys bearing different amounts of nanoparticles.

### 3.5. Ultimate Tensile Strength (UTS)

Figure 9 presents the measured UTS of the wires without and with solution treatment. As expected, the diboride nanoparticles raised the UTS of the wires without solution treatment, which was also the case in previous studies with other nanoparticles [17–19]. Further, the UTS registered in the commercial alloys ranged from 124 MPa to 205 MPa, depending on the Mg content [23]. This behavior can be explained by the amount of cold work that led to significant gains in UTS in these aluminum alloys [25].

Conversely, the measured UTS of the solution-treated wires declined slightly for higher amounts of nanoparticles. This behavior agrees with Figure 8 and reveals how the signal amplitude diminishes for larger amounts of nanoparticles in solution-treated specimens. In other words, the higher the content of nanoparticles, the lower the number of precipitating Mg atoms, and, therefore, the lower the UTS results. According to Härtel et al., whom we already cited, the presence of hard micro-sized particles in a supersaturated matrix produces strain fields that increase the dislocation pinning, and even interactions among them, to favor dynamic strain aging [24]. However, such a strain field enveloping a nanoparticle will not interact with neighboring fields. The dislocation pinning effect would, therefore, be minor and the metastable precipitation, less significant. This is an important finding if one desires to control the mechanical strength by only adding nanoparticles. Because of their size, the $NbB_2$ nanoparticles and the metastable precipitates can only be detected via high-resolution transmission electron microscopy (HRTEM) (not available to the authors). In particular, the nanoparticles could have been extremely hard to find, as their dispersion in the solid matrix (after inoculation) is large. One possible solution (that is not available to the authors either) would be using an *in situ* screw-driven micro tensile stage in a high resolution field emission scanning electron microscope (SEM); that could detect slight submicron changes on the tensile test specimen surface as a result of the PLC phenomenon [26,27].

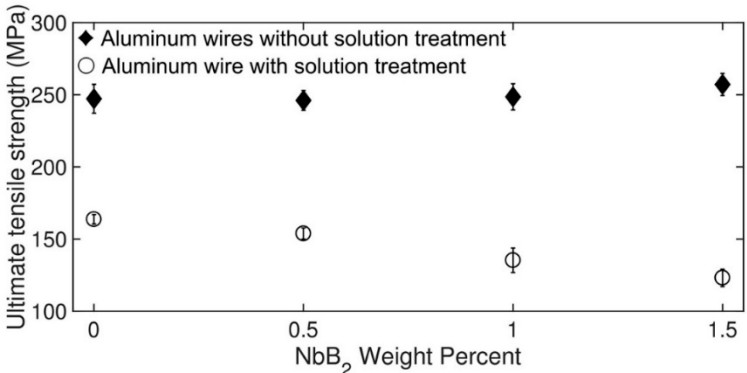

**Figure 9.** Measured ultimate tensile strength of the aluminum wires with and without solution treatment as a function of the content of nanoparticles added.

### 3.6. Statistical Analysis of the PLC Phenomenon

To further validate the lessened amplitude of the PLC phenomenon caused by the addition of the $Al/NbB_2$ nanocomposite, we completed a statistical analysis based on the stress serration amplitude (computed using MATLAB™) in the stress-time curve. For each peak, we first obtained the maximum and minimum values of stress at each time interval (Figure 10).

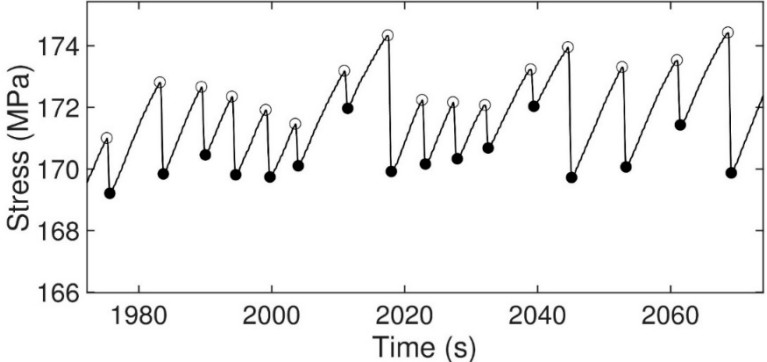

**Figure 10.** Determination of maximum and minimum peaks.

Then, the values for the minimum peaks were subtracted from the values of the maximum peaks. The results correspond to the amplitude of each peak signal.

The measured stress amplitude shows a left-skewed distribution (Figure 11). Therefore, the data were transformed to render a normal distribution, using Minitab™ (Figure 12). Equation (1) shows the power function that rendered a normal distribution, where TA is transformed amplitude and A is the amplitude.

$$TA = A^\lambda \tag{1}$$

The lambda ($\lambda$) value is obtained by iteration in Minitab™. A value of $\lambda = 0.19$ yields a normal distribution (Figure 12). Figure 13 depicts the transformed amplitude as a function of the nanoparticle content.

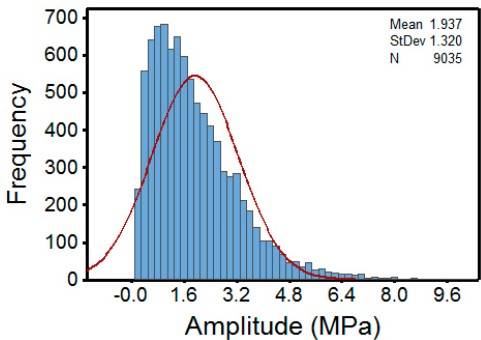

**Figure 11.** Histogram of the amplitude of the peak.

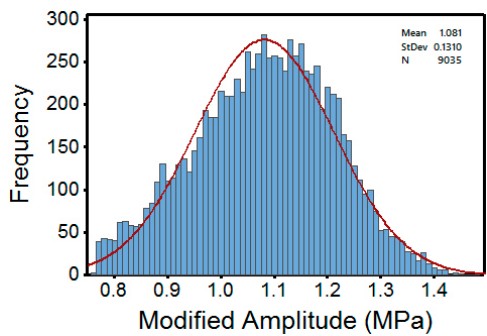

**Figure 12.** Histogram of the transformed amplitude of the peak.

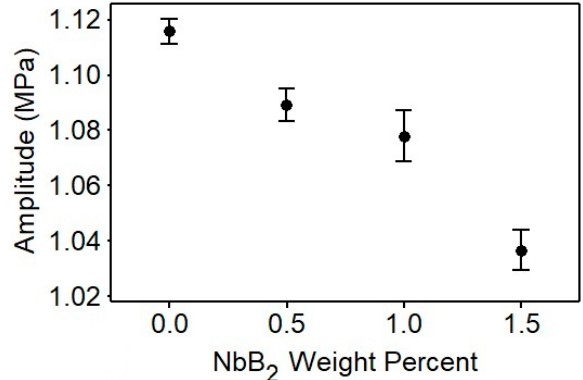

**Figure 13.** The amplitude of the serrated signal upon PLC.

At this point, one must underscore that due to the expected scattering of the results (produced by the serration), thorough statistical analysis helped analyze the results more effectively. Thus, Table 1 provides the resulting analysis of variance (ANOVA) obtained using Minitab™ displaying the degrees of freedom (DF), sum of squares (SS), mean sum of squares (MS), F-statistic (that determines whether the regression is statistically significant), and *p*-value, which allowed determining whether to reject the null hypothesis. For a significant level of $\alpha = 0.05$, the *p*-values are less than 0.05, which indicates that the NbB$_2$ content was affecting considerably, the signal amplitude.

**Table 1.** ANOVA: Amplitude of the peak (MPa) and %NbB$_2$.

| Source | DF | SS | MS | F | *p*-Value |
|---|---|---|---|---|---|
| Samples | 3 | 519.8 | 173.278 | 102.78 | 0.000 |

Again, the ANOVA results confirmed that the content of nanoparticles decreases the amplitude of the PLC phenomenon. This behavior can be explained by the interactions between the dislocations with the nanoparticles, which hampers the movement of those dislocations [2,28].

In closing, the present research provides an alternative means to control the PLC phenomenon via the addition of a nanocomposite material to an Al-Mg alloy while concurrently offering a means to analyze and quantify the serrated flow. With the increasing application of new joining technologies for aluminum, including friction stir processing [29,30] and alternative welding materials [31], the manifestation of PLC in the base metal, the heat affected zone, or the weld requires control. In this context, the present work proposes an appealing approach to counteract DSA upon such fabrication processes using an Al/NbB$_2$ nanocomposite.

## 4. Conclusions

The results and ensuing analysis of the addition of an Al/MgB$_2$ nanocomposite to Al-Mg alloys subject to dynamic strain aging led to the following conclusions:

- In the present study, the Portevin-Le Chatelier effect is present in all treated Al-Mg alloys containing NbB$_2$ nanoparticles.
- Naturally, the phenomenon is absent in specimens without the solution treatment. This was used as baseline for comparison purposes.
- A MATLAB™ code, developed using the Fourier transform, permitted quantification of the PLC signal.
- As expected, the NbB$_2$ nanoparticles raised the ultimate tensile strength of the wires.
- The most important conclusion indicates that the nanoparticle's addition attenuates the serration effect in solution-treated Al-Mg specimens.
- The results were validated via an analysis of variance of the stress amplitude as a function of the level of nanoparticles added.

**Author Contributions:** D.F.-A. and M.M.-G. manufactured the wires, analyzed and interpreted the data obtained from the characterization of those wires, and wrote a significant portion of the manuscript. J.J.M.-S., D.F.-A., and L.M.V. developed the MATLAB™ code. O.M.S. leads the Nanotechnology Center hosting this research and contributed to the manuscript preparation.

**Funding:** This material is based upon work supported by the US National Science Foundation under grant No. HRD 1345156 (CREST program).

**Acknowledgments:** The authors would like to thank the Materials Research Laboratory technician, Boris Rentería for his assistance in the completion of this research. The tensile test machine was acquired through a grant provided by the Solid Waste Management Authority of Puerto Rico.

**Conflicts of Interest:** The authors declare no conflict of interest.

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
