# Peer review of "Al/Niobium Diboride Nanocomposite’s Effect on the Portevin-Le Chatelier Phenomenon in Al-Mg Alloys"

_jcs, doi:10.3390/jcs3030070_

Round 1
Reviewer 1 Report
This paper reprots the effect of NiB2 on the PLC of AlMg alloy. It will be helpful to control the PLC of AlMg alloy. One comment is that the authors claim that "the higher the content of nanoparticles, the lower the number of precipitating Mg atoms, and, therefore, a lower UTS results" (line 167-169), it would be helpful to show the precipitates with the presence of differen contents of NbB2. Some more experimental support (microstructure) is needed.
Author Response
Comments and Suggestions for Authors
This paper reprots the effect of NbB2 on the PLC of AlMg alloy. It will be helpful to control the PLC of AlMg alloy. One comment is that the authors claim that "the higher the content of nanoparticles, the lower the number of precipitating Mg atoms, and, therefore, a lower UTS results" (line 167-169), it would be helpful to show the precipitates with the presence of differen contents of NbB2. Some more experimental support (microstructure) is needed.
R/ Although we certainly agree with the reviewer on extending this research in view of the relevant findings, this first work on the addition of NbB2 to solid-solutioned Al-Mg alloys (the first of its kind) has only opened the door for further experimentation. As stated in the manuscript, the ultimate goal would be to control the PLC phenomenon via exact dosing of the alloy with nanoparticles. This falls beyond the scope of the present work.
Moreover, based on the indirect insight provided by the Matlab analysis, our claim is that Mg atoms remain in the Al solid solution when the nanoparticles are present as NbB2 nanoparticles inhibit the precipitation. We intended to explore the effect of the amount of NbB2, not to establish a complete mapping of that effect for a large range of nanoparticles added. Due to the complex production of the nanocomposite pellets used to inoculate the Al-Mg melt, said analysis is left for our next manuscript. As stated, this is the first report on the aforementioned effect.
Furthermore, NbB2 nanoparticles and the metastable precipitates cannot be detected via regular microstructure analysis. Only an HRTEM (not available to the authors) could have barely distinguished the nanoparticles, as their dispersion in the solid matrix (after inoculation) is large. One possible solution (that is not available to the authors either) would be using an in situ screw-driven micro tensile stage in a high resolution field emission SEM that could detect slight submicron changes on the tensile test specimen surface as a result of the PLC phenomenon [1][2]. We commented on this in the manuscript, as a suggestion for those willing to continue exploring the dynamic strain again based on our discovery (Lines 169-183)
References
[1] B. Wisner and A. Kontsos, “Investigation of particle fracture during fatigue of aluminum 2024,” Int. J. Fatigue, vol. 111, pp. 33–43, 2018.
[2] K. Perzynski, J. Wang, K. Radwanski, K. Muszka, and L. Madej, “Identification of critical strains for the random cellular automata finite element failure model based on in-situ tensile test,” Mech. Mater., vol. 133, pp. 154–164, 2019.
Reviewer 2 Report
The paper is focused on the Al/Niobium Diboride Nanocomposite’s Effect on the Portevin-Le Chatelier Phenomenon in Al-Mg Alloys. The MS can be published after the following revisions:
- Some examples of stress vs strain curves should be presented.
- I suggest to determine the energy stored during the tensile test by the determination of the area under the stress vs strain curves as showed in Nanomaterials 2017, 7(8), 199.
Author Response
Comments and Suggestions for Authors
The paper is focused on the Al/Niobium Diboride Nanocomposite’s Effect on the Portevin-Le Chatelier Phenomenon in Al-Mg Alloys. The MS can be published after the following revisions:
- Some examples of stress vs strain curves should be presented.
R/ We added a graph with an example of the resulting tensile curves of the aluminum wires with and without solution treatment. The PLC phenomenon is apparent in the solution-treated specimen.
- I suggest to determine the energy stored during the tensile test by the determination of the area under the stress vs strain curves as showed in Nanomaterials 2017, 7(8), 199.
R/ We appreciate the published article recommendation; however, said reference does not provide a detailed explanation of how the authors computed the area associated with said energy. One must keep in mind that in our case there is an artifact caused by the tensile machine noise. This was corrected in our work, affecting the stress-strain curve. Being said that, at this moment, we are writing a more elaborate MatLab™ code to determine the energy released that will be included in future publications when the code is ready and validated with additional experimental data. Nonetheless, we proceeded as follows: To determine the energy released due to the PLC phenomenon, we obtained the maximum peaks of the PLC signal. With the maximum peaks, a linear regression was generated (Figure 1). Then, we computed the area under the curve and subtracted the area under the red line (stress-strain curve); thus, the resulting (shaded) area corresponds to the energy released as a result of the PLC.
Figure 1. Assessment of the energy released as a result of the PLC phenomenon.

Round 2
Reviewer 1 Report
The revised paper has been improved. It is recommented to be published in the present form.